# BALAD-2 Emerges as the Most Accurate Prognostic Model in Hepatocellular Carcinoma: Results from a Biobank-Based Cohort Study

**DOI:** 10.3390/cancers17213457

**Published:** 2025-10-28

**Authors:** Coskun Ozer Demirtas, Fatih Eren, Demet Yilmaz Karadag, Yasemin Kaldirim Armutcuoglu, Tugba Tolu, Javid Huseyinov, Ugur Ciftci, Tuba Yilmaz, Sehnaz Akin, Feyza Dilber, Osman Cavit Ozdogan

**Affiliations:** 1Division of Gastroenterology and Hepatology, School of Medicine, Marmara University, 34899 Istanbul, Türkiye; ykaldirim@yahoo.com (Y.K.A.); tugbatolu139@gmail.com (T.T.); cavidmurad1793@gmail.com (J.H.); ugurdr37@gmail.com (U.C.); tuba_yilmaz83@hotmail.com (T.Y.); drfgunduz@yahoo.com (F.D.); osmanozdogan@yahoo.com (O.C.O.); 2Institute of Gastroenterology, Marmara University, 34854 Istanbul, Türkiye; erenfatih78@yahoo.com (F.E.); yilmz.demet@gmail.com (D.Y.K.); 3Section of Digestive Diseases, Yale University, New Haven, CO 06511, USA; 4Department of Medical Biology, School of Medicine, Marmara University, 34854 Istanbul, Türkiye; 5Faculty of Medicine, Recep Tayyip Erdogan University, 53200 Rize, Türkiye; 6Department of Internal Medicine, School of Medicine, Marmara University, 34854 Istanbul, Türkiye; sehnazakin96@gmail.com

**Keywords:** Alfa-fetoprotein, Alfa-fetoprotein-L3, Des-gamma-carboxy prothrombin, hepatocellular carcinoma, BALAD-2

## Abstract

Accurate prediction of survival in patients with hepatocellular carcinoma (HCC) is important for multidisciplinary decision-making and follow-up. In this study, we compared several blood-based biomarkers and scoring systems, including AFP; AFP-L3%; DCP; and models such as GALAD, BALAD, BALAD-2, GAAP, ASAP, the Doylestown algorithm, and aMAP. Using data from 186 patients with HCC, we found that all biomarkers and models were related to survival. Among them, the BALAD-2 score provided the best and most consistent performance, particularly in patients with viral liver disease and those receiving curative treatments. These results suggest that BALAD-2 could be a valuable tool for risk assessment and treatment planning in HCC.

## 1. Introduction

Hepatocellular carcinoma (HCC) is the fifth most prevalent malignancy and the third leading cause of cancer-related mortality worldwide, with incidence and death rates projected to increase by over 50% by 2040 [1,2]. Despite advances in diagnostics and therapeutics, prognostication of HCC remains challenging owing to its heterogeneous nature. Unlike other solid malignancies, HCC prognosis depends not only on the tumor burden but also on three other key factors: hepatic synthetic function, overall health status of the patient, and type of treatment. HCC encompasses heterogeneous subgroups that show differences in tumor burden and liver function and is associated with wide differences in applied treatment modalities and survival outcomes [3]. Staging of HCC is a crucial step in determining the management strategy and thereby prognosis. Barcelona Clinic Liver Cancer (BCLC) is the most widely used staging system for treatment allocation and prognostication [4].

An accurate prognostic assessment is essential for guiding therapeutic decisions, predicting outcomes, and stratifying patients in clinical trials. Traditional prognostic systems such as BCLC and liver function scores, such as Child–Pugh (CP), Model for End-Stage Liver Disease (MELD), and Albumin-bilirubin (ALBI), are widely used for prognosis determination in clinical practice and trials. However, these models may not adequately account for tumor biology or serum biomarkers that reflect tumor aggressiveness. Alpha-fetoprotein (AFP) has long been used as an additional tool for predicting prognostic outcomes. In addition to AFP, lens culnaris agglutinin-reactive AFP (AFP-L3%) and des-gamma-carboxy prothrombin (DCP) have long been investigated for their diagnostic and prognostic utility in HCC, but their individual performances are variable [5,6,7,8].

Several composite models incorporating these biomarkers have been developed to enhance prognostic precision. The GALAD score, which combines age, sex, AFP, AFP-L3%, and DCP, was originally designed to improve early HCC detection [9,10,11,12]. The GAAP and ASAP models, which exclude AFP-L3% and rely on fewer biomarkers (AFP, DCP, age, and sex), offer comparable performance and have also been validated for surveillance purposes [13,14]. Other clinical scoring systems, such as the aMAP score (based on age, sex, albumin, bilirubin, and platelets) and the Doylestown algorithm (including age, sex, AFP, alanine transaminase [ALT], and alkaline phosphatase [ALP]), provide additional approaches to HCC risk stratification [15,16]. The BALAD and BALAD-2 models, specifically developed for prognostication, incorporate both tumor biomarkers (AFP, AFP-L3%, and DCP) and liver function indicators (albumin and bilirubin). The original BALAD score assigns binary values to these five parameters, generating a score from 0 to 5, based on clinical thresholds [17]. BALAD-2 improves upon this by incorporating the same variables as continuous values in a regression-based model with the aim of enhancing discriminatory power [18]. Although several biomarker-based models such as GALAD, GAAP, ASAP, aMAP and the Doylestown algorithm were originally developed for early detection or HCC risk stratification, emerging evidence indicates that these models also carry prognostic information related to tumor stage, recurrence, and overall survival [19,20,21,22,23]. Given their increasing use in clinical and research settings, a direct head-to-head comparison with established prognostic models such as BALAD and BALAD-2 is warranted to determine their relative predictive accuracy and potential applicability in survival estimation.

Although biomarker-based models have shown promise in selected cohorts, direct comparisons of their prognostic performance in well-characterized, real-world HCC populations remain limited. Therefore, in this study, we evaluated the prognostic utility of individual serum biomarkers (AFP, AFP-L3%, and DCP) and seven composite models (GALAD, GAAP, ASAP, BALAD, BALAD-2, Doylestown algorithm, and aMAP) using data from a biobank-based cohort of patients with HCC. We further assessed their performance across different subgroups based on the treatment intent and underlying liver disease etiology.

## 2. Materials and Methods

### 2.1. Study Design and Patient Population

This retrospective cohort study was conducted at Marmara University, Institute of Gastroenterology, Istanbul, Türkiye. Patients with a newly diagnosed, treatment-naive HCC between January 2019 and January 2024 were included in the analyses. Eligible participants were adults (>18 years) with a diagnosis of HCC established either by histologic confirmation or characteristic imaging features according to the European Association for the Study of the Liver (EASL) guidelines with complete clinical, laboratory, and imaging data available [24]. Exclusion criteria included a diagnosis of combined HCC-cholangiocarcinoma or other non-HCC primary liver malignancies, liver transplantation at or before the time of HCC diagnosis, loss to follow-up, or missing key data. All patients were enrolled from the Marmara University HCC Biobank, which includes prospectively collected serum samples and clinical data from patients with liver diseases.

### 2.2. Data Collection

Demographic data (age, sex, height, weight, and body mass index [BMI]) were collected for all participants. Comorbidities such as diabetes mellitus, hypertension, and hyperlipidemia were determined based on clinical history or laboratory findings. The underlying etiology of liver disease—hepatitis B virus (HBV), hepatitis C virus (HCV), metabolic-associated steatotic liver disease (MASLD), or alcohol-related liver disease (ALD) —was identified. HBV etiology was defined by the presence of hepatitis B surface antigen (HbsAg) positivity, while HCV etiology was defined by the presence of hepatitis C virus antibody (anti-HCV) and/or detectable HCV-RNA. Patients with MASLD were identified based on radiologic evidence of hepatic steatosis in the absence of significant alcohol consumption, negativity for both HbsAg and anti-HCV antibodies, and exclusion of other causes of chronic liver disease. In cases where hepatic steatosis co-existed with HbsAg or anti-HCV positivity, the viral etiology was prioritized over MASLD to avoid overlap. Laboratory parameters including albumin, bilirubin, sodium, and international normalized ratio (INR) were obtained. The baseline tumor characteristics and staging were recorded. Tumor characteristics were assessed using contrast-enhanced imaging at the time of diagnosis. Tumor staging was performed using the BCLC system [4].

Treatment modalities were classified as curative (surgical resection, radiofrequency ablation [RFA]), non-curative (transarterial chemoembolization [TACE], transarterial radioembolization [TARE], systemic therapy), or best supportive care (BSC) [4,24].

### 2.3. Serum Biomarker Measurements and Calculation of Scoring Models

Serum samples were collected from each participant, stored at –80 °C, and subsequently analyzed for the concentrations of AFP, AFP-L3, and DCP using an automated immunoassay system assay on the μTASWako i30 immuno-analyzer (Wako Chemicals, Neuss, Germany) [25]. The analytical assay sensitivities were 0.3 ng/mL for AFP and 0.1 ng/mL for DCP. AFP-L3 was reported as a percentage of the total AFP when AFP-L1 and AFP-L3 were 0.3 ng/mL or more. All assays were performed on the same sample in an external laboratory blinded to the clinical data.

All models were calculated for each participant using their original formulas as follows:

GALAD formula [9]: −10.08 + 1.67 × [Sex (1 for male, 0 for female)] + 0.09 × [Age] + 0.04 × [AFP-L3%] + 2.34 × log_10_[AFP] + 1.33 × log_10_[DCP]

GAAP formula [13]: −11.203 + 0.699 × [sex (1 for male, 0 for female)] + 0.094 × [age] + 1.076 × log_10_[AFP] + 2.376 × log_10_[DCP]

ASAP formula [14]: −7.57711770 + 0.04666357 × [Age] – 0.57611693 × [Sex (1 for female, 0 for male)] + 0.42243533 × ln[AFP] + 1.10518910 × ln[DCP].

BALAD-2 formula [18]: 0.02 × (AFP-2.57) + 0.012 × ([AFP-L3]–14.19) + 0.19 × (ln[DCP]−1.93) + 0.17 × ([bilirubin]1/2−4.50) – 0.09 × (albumin35.11)

aMAP formula [15]: ((0.06 × age + 0.89 × sex (male: 1, female: 0) + 0.48 × ((log10 total bilirubin × 0.66) + (albumin × −0.085)) −0.01 × platelets) + 7.4)/14.77 × 100

Doylestown algorithm [16]: 1/(1 + EXP (−(−10.307 + (0.097 × age) + (1.645 × gender) + (2.314 × logAFP) + (0.011 × ALP) + (−0.008 × ALT)))).

### 2.4. Statistical Analysis

Descriptive statistics were used to summarize baseline characteristics. The primary endpoint was overall survival (OS), defined as the time from HCC diagnosis to death or the last follow-up. Univariate and multivariate Cox proportional hazards regression analyses were performed to identify the variables associated with OS. Univariate Cox regression analyses were initially conducted for all variables, including each biomarker and the composite model. Variables demonstrating statistical significance in the univariate analyses were subsequently included in multiple multivariable logistic regression models to identify independent associations with OS. To prevent multicollinearity, each multivariable model incorporated only one biomarker or composite model alongside covariates that did not overlap with the components of the included biomarker or model. Adjusted hazard ratios (HRs) with 95% confidence intervals (Cis), beta coefficients, and *p*-values were reported for all regression models.

The discriminative performance of Individual biomarkers and prognostic models was assessed using the time-dependent area under the receiver operating characteristic curve (AUROC) at 1-, 2-, 3-, and 5-year intervals and Harrel’s concordance index (c-index). Although the primary goal was to compare the biomarkers and composite models, liver function scores (ALBI, Child–Pugh, and MELD) and BCLC staging system were also evaluated for prognostic utility to demonstrate a comparison with the individual biomarkers and composite models. Subgroup analyses were conducted based on the treatment intent (curative vs. non-curative) and liver disease etiology (viral vs. non-viral). Statistical significance was defined as a two-sided *p*-value of <0.05.

### 2.5. Ethical Considerations

This study was conducted in compliance with the Declaration of Helsinki and its subsequent amendments, with ethical approval obtained under the Ethics Committee Protocol Number 28.06.2024.769, dated 08.07.2024. Informed consent was obtained from all participants at the time of biobank enrollment (Ethics Committee Protocol Number 09.2019.716, dated 13 September 2019). All data and serum samples were obtained from the HCC Biobank of Marmara University.

## 3. Results

### 3.1. Baseline Characteristics

A total of 186 patients with hepatocellular carcinoma (HCC) were included in this study. The baseline demographics and laboratory characteristics are summarized in Table 1. The median age was 65 (37–88) years, and 74.7% of the patients were male. The most common underlying etiology was HBV (49.5%), followed by MASLD (29.0%), and HCV (14.0%). Ascites and esophageal varices were present in 34.9% and 40.3% of the patients, respectively.

Tumor characteristics, serum biomarkers, and prognostic model values are shown in Table 2. The median tumor size was 40 (10–162) mm, and 60.8% of the patients presented with a single lesion. PVT and vascular invasion were present in 10.8% and 17.2% of the cases, respectively. In terms of tumor stage, 45.2% had early stage HCC (BCLC 0 or A), 37.1% had BCLC-B, 12.9% had BCLC-C, and 4.8% had BCLC-D stage HCC. The treatments included resection (10.2%), RFA (28.0%), TACE (32.8%), TARE (5.9%), systemic therapy (3.8%), and BSC (18.3%).

### 3.2. Prognostic Impact of Biomarkers and Models

Univariate analyses showed that elevated AFP, AFP-L3, and DCP, and all composite models (GALAD, ASAP, GAAP, BALAD, BALAD-2, Doylestown, and aMAP) were significantly associated with poorer survival (all *p* < 0.05), in addition to CTP score, history of variceal bleeding, AST, ALT, ALP, GGT, PVT, extrahepatic metastasis, and treatment category (Appendix A). Kaplan–Meier analyses demonstrated significant survival stratification by individual biomarkers (Figure 1) and composite models (Figure 2). In multivariate analyses, all individual biomarkers and composite models remained independently associated with OS, except the Doylestown and aMAP models, after adjusting for covariates that demonstrated statistical significance in the univariate analyses (*p* < 0.05), including liver function indicators (Child–Pugh), history of variceal bleeding, laboratory (ALT, ALP, sodium), tumor burden (tumor size, number of lesions, presence of portal vein thrombosis, extrahepatic metastasis), and treatment category (curative, non-curative, or BSC) variables (Table 3).

### 3.3. Model Discrimination and Subgroup Performance

The prognostic performance of individual biomarkers and models in the entire cohort, as well as in the etiology-based subgroups, is presented in Table 4. Among the individual biomarkers, DCP had the highest C-index and AUROCs across all time points. BALAD-2 showed the strongest overall prognostic discrimination, with the highest c-index (0.737) and superior AUROCs in 1st-year (0.827), 2nd-year (0.846), 3rd-year (0.781), and 5th-year (0.716) years. The ASAP and GAAP also performed well in terms of overall prognostic discriminative performance, with c-indices of 0.722 and 0.720, respectively. When stratified by etiology, BALAD-2 showed the best performance in viral-related HCC, with the highest c-index (0.726), 1st-year (0.846), 2nd-year (0.853), and 5th-year (0.758) AUROCs. ALBI slightly outperformed BALAD-2 in the 3rd-year AUROC (0.800 vs. 0.767). In nonviral HCC, GAAP (0.778), ASAP (0.774), Doylestown (0.770), and GALAD (0.764) marginally outperformed BALAD-2 (0.762). Doylestown showed the highest 1st-year (0.807), BCLC led at the 2nd-year (0.837), GAAP at the 3rd-year (0.850), and BALAD had the highest 5th-year (0.733) AUROCs in this group.

The prognostic performance of the individual biomarkers and models stratified by treatment category is shown in Table 5. In the curative treatment subgroup (n = 71), BALAD-2 showed the highest 1st-year (0.865), 2nd-year (0.813), and 5th-year (0.697) AUROCs, and the highest c-index (0.698). At 3rd year, the ALBI (0.770) and aMAP (0.729) slightly outperformed BALAD-2 (0.709). In the non-curative treatment subgroup (n = 81), BALAD-2 again demonstrated the highest 2nd-year (0.829), 3rd-year (0.784) AUROCs and c-index (0.716), while ASAP had the highest 1st-year AUROC (0.799), followed by BALAD-2 (0.782), and DCP led to the 5th-year AUROC (0.744).

## 4. Discussion

In the present study, we evaluated the prognostic performance of individual serum tumor biomarkers and composite models in a well-characterized cohort of patients with HCC. Our findings demonstrate that the BALAD-2 model consistently outperformed both individual biomarkers, liver function scores, BCLC stage, and other composite models in predicting OS regardless of treatment status, particularly among patients with viral-related liver disease. All three biomarkers (AFP, AFP-L3, and DCP) were independently associated with OS. Among these, DCP exhibited the highest c-index and time-dependent AUROCs across multiple time points. This aligns with prior studies that have emphasized its association with vascular invasion, tumor burden, and early recurrence after treatment [26,27]. However, the combined use of these biomarkers in structured models markedly improved prognostic performance compared to their individual use. These findings support the integration of biomarker-based models into the routine clinical risk stratification for HCC.

Among all the models assessed, BALAD-2 demonstrated the highest overall discriminative ability, reflected by a c-index of 0.737 and consistently high AUROCs at the 1-, 2-, 3-, and 5-year intervals. The BALAD-2 model incorporates continuous values of AFP, AFP-L3%, DCP, albumin, and bilirubin, providing a nuanced risk assessment that reflects both tumor biology and hepatic functional reserve. The initial BALAD model, introduced in 2006, consisted of the same variables and demonstrated good discriminatory ability for OS among all patients with HCC, ranging from those with early to advanced stage, and from those who underwent curative treatment to those who underwent non-curative treatment or were followed up with BSC [17]. The refined BALAD-2 model, which was developed in 2014, has been validated and showed superior prognostic ability compared with the original model. It improves upon the original BALAD score by avoiding binary thresholds and reducing information loss, which likely contributes to its superior performance [28,29,30]. A recent study demonstrated that the BALAD-2 serological prognostic model, which was originally developed to analyze the overall HCC population, retained its prognostic ability to predict survival even in advanced unresectable HCCs treated with systemic therapy [31]. This finding further supports its use in clinical trials. While these studies demonstrated the strong prognostic performance of BALAD-2, other studies demonstrated the prognostic utility of GALAD, ASAP, and aMAP models in HCC in comparison to single individual biomarkers [19,20,21,22,23]. To the best of our knowledge, no study has compared the prognostic performances of these composite models in the same study cohort. Therefore, our study has demonstrated, for the first time, that the BALAD-2 score outperforms other available models in predicting overall survival outcomes in an unselected, real-world HCC population.

Subgroup analyses by liver disease etiology revealed that the BALAD-2 model performed particularly well in viral-related HCC, which is consistent with prior validation studies conducted in HBV- and HCV-endemic populations [28,29,30]. In contrast, in non-viral etiology, models such as GAAP, ASAP, Doylestown, and GALAD slightly outperformed BALAD-2, suggesting that the predictive weight of certain biomarkers may differ according to the underlying liver disease [32]. This observation highlights the need for etiologically tailored risk models or adjustments in biomarker interpretation, depending on the disease context.

To homogenize the impact of treatment on prognostic outcomes, we performed a subgroup analysis based on treatment status categorized as curative and non-curative intent. In curatively treated patients, BALAD-2 showed the highest prognostic accuracy at most time points, suggesting its potential utility in preoperative risk assessment and post-treatment surveillance planning. Among patients receiving non-curative therapies, BALAD-2 again ranked highest in most survival endpoints, although ASAP showed a slightly higher performance in the first-year survival prediction. Similarly to BALAD-2, the ASAP model integrates both biomarkers (AFP and DCP, without AFP-L3) and clinical data, but differs in weighting and variable inclusion. Overall, the consistent performance of the BALAD-2 across different treatment modalities suggests its adaptability and robustness in various clinical and treatment settings. Although accurate survival prediction is essential for clinical decision-making, our findings are primarily intended to inform risk stratification rather than to serve as stand-alone treatment selection criteria. The subgroup analyses by treatment intent were exploratory and aimed to assess consistency of model performance rather than to provide treatment-specific guidance. In clinical practice, such models can complement established staging systems (e.g., BCLC) by providing individualized survival estimates that aid multidisciplinary discussion and follow-up planning.

Our study had several strengths, including the use of prospectively collected biobank data and a comprehensive comparative evaluation of both established and emerging prognostic models. However, this is not without its limitations. Despite being a countrywide referral center for HCC, the single-center design may limit the generalizability of our findings. Another limitation is that none of the patients in this cohort received immune checkpoint inhibitor-based systemic therapy, as such regimens were not yet widely available or reimbursed in Türkiye during the study period (2019–2024). Future studies should include patients treated with modern immunotherapy combinations to further validate biomarker-based prognostic models in this therapeutic context. While previous studies have proposed combined indices such as LAD, S-LAD, which integrates tumor diameter with the BALAD model to improve post-transplant and post-hepatectomy prognostication, our study did not focus on therapy specific prognostic models, but rather on models applicable across all stages of HCC [33,34,35]. Future research should evaluate the prognostic performance of BALAD-based models incorporating tumor burden metrics not only in transplant and resection cohort but also in non-curative settings. Although biomarker assays are standardized, inter-assay variability may affect their applicability in external settings. Furthermore, while time-dependent AUROCs and the c-index provide robust estimates of discriminative ability, external validation in independent cohorts is essential to confirm the utility of these models across broader populations.

In conclusion, our results demonstrate that biomarker-based models, particularly BALAD-2, offer superior prognostic accuracy compared with traditional BCLC staging, individual biomarkers (AFP, AFP-L3%, and DCP), and liver function scores (Child–Pugh, MELD, and ALBI) in patients with HCC. These findings support the routine incorporation of validated composite models using AFP, AFP-L3%, and DCP levels in combination with clinical and laboratory data into clinical practice to improve individualized prognostic risk stratification, treatment planning, and long-term monitoring. Future research should focus on refining these models for nonviral HCC and integrating them with molecular and imaging biomarkers for more precise prognostication.

## 5. Conclusions

BALAD-2 demonstrated consistent and robust prognostic performance compared with other biomarker-based and clinical models across different patient subgroups, particularly among those receiving curative therapy and viral etiologies. Among the individual biomarkers, DCP exhibited the highest c-index and AUROCs at all evaluated time points. BALAD-2 provided the strongest overall prognostic discrimination, while the ASAP and GAAP also showed favorable performance. Collectively, these findings support the integration of biomarker-based prognostic models—especially BALAD-2—into clinical risk stratification and multidisciplinary decision-making in HCC management.

## Figures and Tables

**Figure 1 cancers-17-03457-f001:**
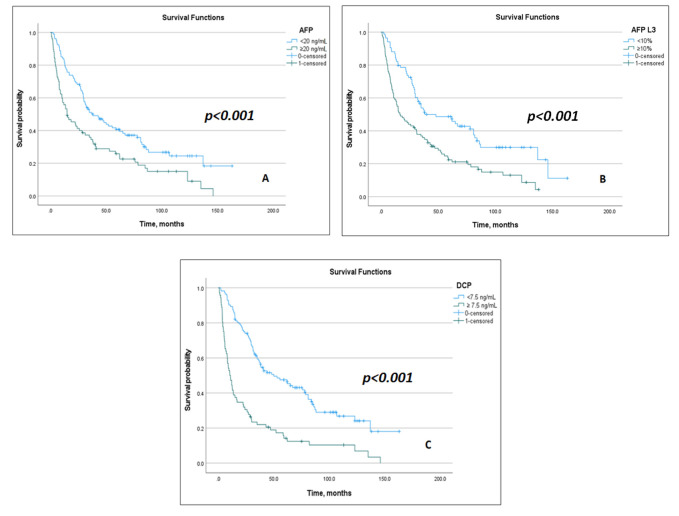
Kaplan–Meier Survival Curves for Each Individual Biomarker. (**A**): Alpha-fetoprotein (AFP); (**B**): Alpha-fetoprotein-L3 (AFP-L3); (**C**): Des-gamma-carboxy prothrombin (DCP).

**Figure 2 cancers-17-03457-f002:**
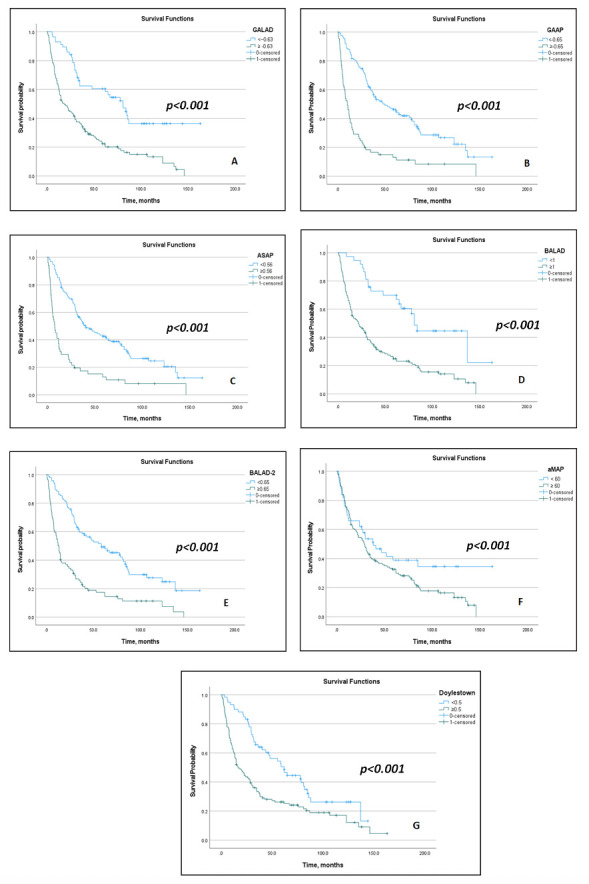
Kaplan–Meier Survival Curves for Each Composite Model. (**A**): GALAD; (**B**): GAAP; (**C**): ASAP; (**D**): BALAD; (**E**): BALAD-2; (**F**): aMAP; (**G**): Doylestown.

**Table 1 cancers-17-03457-t001:** Baseline demographics and laboratory characteristics.

	HCC (n = 186)
Age, median (IQR), years	65(37–88)
Gender, n (%) - Male - Female	139 (74.7) 47 (25.3)
Body-mass index, median (IQR), kg/m ^2^	28.3 (16.5–45.7)
Diabetes mellitus, n (%)	73 (39.2)
Hypertension, n (%)	63 (33.9)
Hyperlipidemia, n (%)	26 (14.0)
Etiology, n (%) - HBV - MASLD-cryptogenic - HCV - ALD - Autoimmune—PBC - HBV+HDV - HBV+HCV	92 (49.5) 54 (29.0) 26 (14.0) 3 (1.6) 3 (1.6) 5 (2.7) 3 (1.6)
Ascites, n (%)	65 (34.9)
Varices, n (%)	75 (40.3)
Variceal bleeding history, n (%)	17 (9.1)
CTP score, median (IQR)	6 (5–12)
MELD score, median (IQR)	10 (6–28)
ALBI score, median (IQR)	−2.41 (−4.39–1.03)
Albumin, median (IQR), gr/dL	3.8 (2.1–6.4)
INR, median (IQR)	1.2 (0.9–3.5)
Total bilirubin, median (IQR), mg/dL	1.1 (0.2–12.8)
Alanine aminotransferase, median (IQR), U/L	37 (10–258)
Alkaline phosphatase, median (IQR), U/L	127 (36–642)
Gamma-glutamyl transferase, median (IQR), U/L	84 (11–1612)
Platelet count, median (IQR), ×1000/m^3^	144 (36–838)
Creatinine, median (IQR), mg/dL	0.8 (0.4–4.4)
Sodium, median (IQR), mEq/L	138 (121–145)

AFP: Alfa-fetoprotein; AFP-L3: Lens culinaris agglutinin-reactive Alfa-fetoprotein; ALBI: Albumin-bilirubin; ALD: Alcohol-associated Liver Disease; CTP: Child-Turcotte-Pugh; DCP: des-gamma-carboxy prothrombin; DM: Diabetes mellitus; HBV: Hepatitis B Virus; HCV: Hepatitis C Virus; HE: Hepatic Encephalopathy; MASLD: Metabolic-Associated Steatotic Liver Disease; MELD: Model for End-stage Liver Disease; INR: International normalized ratio; PBC: Primary Biliary Cholangitis.

**Table 2 cancers-17-03457-t002:** Tumor characteristics, serum biomarkers and prognostic models.

	HCC (n = 186)
Tumor size, median (IQR), mm	40 (10–162)
Number of tumors, n (%) • 1 • 2 • 3 • >3 • Diffuse	113 (60.8) 22 (11.8) 17 (9.1) 30 (16.1) 4 (2.2)
ECOG, n (%) • 0 • 1 • 2 • 3	162 (87.2) 21 (11.3) 1 (0.5) 2 (1.0)
Lymph node involvement, n (%)	27 (14.5)
Portal vein thrombosis, n (%)	20 (10.8)
Vascular invasion, n (%)	32 (17.2)
Extrahepatic metastasis, n (%)	7 (3.8)
BCLC stage, n (%) • 0 • A • B • C • D	15 (8.1) 69 (37.1) 69 (37.1) 24 (12.9) 9 (4.8)
HCC treatment, n (%) • Resection • RFA • TACE • TARE • Sorafenib • TACE+RFA • TACE+sorafenib • Best supportive care	19 (10.2) 52 (28.0) 61 (32.8) 11 (5.9) 7 (3.8) 1 (0.5) 1 (0.5) 34 (18.3)
AFP, median (IQR), ng/ml	14.3 (0.7–200,000.0)
AFP-L3, median (IQR), %	12.1 (0.5–94.4)
DCP, median (IQR), ng/mL	3.3 (0.1–14980.0)
GALAD score, median (IQR)	1.44 (−5.53–16.67)
BALAD score, median (IQR)	2.00 (0.00–5.00)
BALAD-2 score, median (IQR)	0.45 (−2.39–5094.05)
ASAP, median (IQR)	−2.11 (−7.22–9.76)
GAAP, median (IQR)	−1.58 (−7.62–10.43)
Doylestown algorithm, median (IQR)	0.73 (0.02–1.00)
aMAP, median (IQR)	63.41 (19.79–80.17)

AFP: Alpha-fetoprotein; AFP-L3: Lens culnaris agglutinin-reactive alpha-fetoprotein; BSC: Best supportive care; DCP: Des-gamma-carboxy prothrombin; BCLC: Barcelona Clinic Liver Cancer; ECOG: Eastern Cooperative Oncology Group scale; HCC: Hepatocellular carcinoma; RFA: Radiofrequency ablation; TACE: Transarteriel chemoembolization; TARE: Transarteriel radioembolization.

**Table 3 cancers-17-03457-t003:** Univariate and Multivariate Cox Regression Analysis of Biomarkers and Composite Models for Overall Survival.

	Univariate	Multivariate			
	*p*-Value	Beta-Coefficient	aHR	95% CI	*p*-Value
AFP	<0.001	0.000	1.000 ^+^	1.000–1.000	<0.001 *
AFP-L3	<0.001	0.010	1.009 ^+^	1.002–1.017	0.015 *
DCP	<0.001	0.000	1.000 ˆ	1.000–1.000	0.002 *
GALAD	<0.001	0.095	1.074 ^#^	1.025–1.126	0.003
ASAP	<0.001	0.160	1.115 ^†^	1.036–1.200	0.004 *
GAAP	<0.001	0.164	1.090 ^†^	1.013–1.173	0.021 *
BALAD	<0.001	0.249	1.307 ^#^	1.093–1.563	0.003 *
BALAD-2	<0.001	0.000	1.000 ^#^	1.000–1.001	0.007 *
DOYLESTOWN	<0.001	0.974	1.605 ^±^	0.732–3.521	0.238
AMAP	0.048	0.242	1.015 ^º^	0.989–1.042	0.264

AFP: Alpha-fetoprotein; AFP-L3: Lens culnaris agglutinin-reactive alpha-fetoprotein; aHR= Adjusted hazard ratio; DCP: Des-gamma-carboxy prothrombin; HR: Hazard ratio. ^+^ Adjusted for Child–Pugh–Turcotte Score, history of variceal bleeding, alanine aminotransferase, alkaline phosphatase, sodium, maximum tumor size, number of tumoral lesions, portal vein thrombosis, extrahepatic metastasis, treatment category, AFP-L3 and DCP. ˆ Adjusted for Child–Pugh–Turcotte Score, history of variceal bleeding, alanine aminotransferase, alkaline phosphatase, sodium, maximum tumor size, number of tumoral lesions, portal vein thrombosis, extrahepatic metastasis, treatment category, AFP and AFP-L3. ^#^ Adjusted for Child–Pugh–Turcotte Score, history of variceal bleeding, alanine aminotransferase, alkaline phosphatase, gamma-glutamyl transferase, sodium, maximum tumor size, number of tumoral lesions, portal vein thrombosis, extrahepatic metastasis and treatment category. ^†^ Adjusted for Child–Pugh–Turcotte Score, history of variceal bleeding, alanine aminotransferase, alkaline phosphatase, sodium, AFP-L3, maximum tumor size, number of tumoral lesions, portal vein thrombosis, extrahepatic metastasis and treatment category. ^±^ Adjusted for Child–Pugh–Turcotte Score, history of variceal bleeding, sodium, maximum tumor size, number of tumoral lesions, portal vein thrombosis, extrahepatic metastasis, treatment category, AFP, AFP-L3 and DCP. ^º^ Adjusted for Child–Pugh–Turcotte Score, history of variceal bleeding, alanine aminotransferase, alkaline phosphatase, sodium, maximum tumor size, number of tumoral lesions, portal vein thrombosis, extrahepatic metastasis, treatment category, AFP, AFP-L3 and DCP. * indicates *p*-values < 0.05.

**Table 4 cancers-17-03457-t004:** Prognostic Performance of Individual Biomarkers and Models, Including Subgroup Analysis by Underlying Liver Disease Etiology.

**Whole cohort (n = 186)**					
	**1st-Year AUROC**	**2nd-Year AUROC**	**3rd-Year AUROC**	**5th-Year AUROC**	**c-İndex**
AFP	0.766 (0.684–0.848)	0.738 (0.664–0.812)	0.677 (0.600–0.755)	0.584 (0.445–0.723)	0.665
AFP-L3	0.741 (0.664–0.818)	0.701 (0.624–0.777)	0.631 (0.550–0.713)	0.593 (0.444–0.742)	0.637
DCP	0.794 (0.721–0.866)	0.781 (0.715–0.847)	0.739 (0.667–0.810)	0.702 (0.565–0.839)	0.703
ALBI	0.731 (0.653–0.809)	0.787 (0.720–0.853)	0.784 (0.714–0.855)	0.608 (0.461–0.755)	0.702
CPS	0.654 (0.563–0.744)	0.690 (0.612–0.769)	0.696 (0.618–0.774)	0.633 (0.473–0.793)	0.634
MELD	0.626 (0.529–0.723)	0.635 (0.553–0.718)	0.626 (0.538–0.703)	0.507 (0.349–0.664)	0.594
GALAD	0.804 (0.730–0.878)	0.801 (0.736–0.866)	0.731 (0.659–0.804)	0.647 (0.502–0.792)	0.709
ASAP	0.820 (0.749–0.892)	0.809 (0.745–0.872)	0.756 (0.685–0.826)	0.660 (0.513–0.807)	0.721
GAAP	0.807 (0.735–0.880)	0.813 (0.751–0.875)	0.761 (0.692–0.831)	0.673 (0.520–0.827)	0.720
BALAD	0.801 (0.732–0.871)	0.799 (0.735–0.863)	0.733 (0.659–0.806)	0.675 (0.539–0.811)	0.705
BALAD-2	0.827 (0.764–0.890)	0.846 (0.791–0.901)	0.781 (0.713–0.850)	0.716 (0.561–0.872)	0.737
DOYLESTOWN	0.789 (0.711–0.868)	0.790 (0.722–0.858)	0.733 (0.660–0.807)	0.648 (0.480–0.816)	0.690
AMAP	0.471 (0.374–0.568)	0.596 (0.510–0.681)	0.605 (0.522–0.687)	0.502 (0.347–0.657)	0.546
BCLC	0.788 (0.719–0.857)	0.749 (0.678–0.820)	0.696 (0.619–0.773)	0.568 (0.405–0.732)	0.677
**Viral etiology ** **(n = 126)**					
	**1st-year AUROC**	**2nd-year AUROC**	**3rd-year AUROC**	**5th-year AUROC**	**c-index**
AFP	0.770 (0.668–0.871)	0.744 (0.654–0.833)	0.647 (0.548–0.745)	0.640 (0.482–0.798)	0.648
AFP-L3	0.752 (0.654–0.850)	0.701 (0.608–0.795)	0.600 (0.496–0.703)	0.594 (0.390–0.798)	0.621
DCP	0.837 (0.760–0.913)	0.793 (0.714–0.872)	0.719 (0.628–0.810)	0.685 (0.543–0.827)	0.683
ALBI	0.757 (0.654–0.859)	0.792 (0.708–0.877)	0.800 (0.719–0.881)	0.638 (0.465–0.812)	0.717
CPS	0.652 (0.535–0.769)	0.686 (0.587–0.786)	0.704 (0.610–0.798)	0.654 (0.475–0.834)	0.633
MELD	0.636 (0.509–0.762)	0.628 (0.524–0.732)	0.610 (0.508–0.712)	0.548 (0.353–0.743)	0.591
GALAD	0.819 (0.730–0.908)	0.803 (0.724–0.882)	0.695 (0.601–0.788)	0.658 (0.504–0.812)	0.685
ASAP	0.845 (0.766–0.924)	0.807 (0.728–0.886)	0.714 (0.632–0.811)	0.644 (0.496–0.792)	0.694
GAAP	0.822 (0.736–0.908)	0.810 (0.732–0.887)	0.721 (0.632–0.811)	0.661 (0.508–0.813)	0.690
BALAD	0.828 (0.751–0.906)	0.802 (0.725–0.879)	0.703 (0.610–0.798)	0.650 (0.469–0.831)	0.692
BALAD-2	0.846 (0.774–0.918)	0.853 (0.788–0.918)	0.767 (0.683–0.852)	0.758 (0.596–0.920)	0.726
DOYLESTOWN	0.770 (0.666-.875)	0.769 (0.681–0.857)	0.668 (0.571–0.764)	0.615 (0.434–0.796)	0.645
AMAP	0.383 (0.267–0.499)	0.536 (0.428–0.645)	0.564 (0.461–0.668)	0.484 (0.295–0.673)	0.519
BCLC	0.790 (0.509–0.762)	0.710 (0.616–0.804)	0.674 (0.578–0.770)	0.559 (0.366–0.751)	0.644
**Non-viral etiology ** **(n = 60)**					
	**1st-year AUROC**	**2nd-year AUROC**	**3rd-year AUROC**	**5th-year AUROC**	**c-index**
AFP	0.741 (0.592–0.890)	0.695 (0.552–0.837)	0.718 (0.585–0.851)	0.493 (0.241–0.744)	0.702
AFP-L3	0.702 (0.563–0.840)	0.676 (0.536–0.816)	0.679 (0.540–0.818)	0.608 (0.455–0.762)	0.673
DCP	0.741 (0.599–0.883)	0.799 (0.683–0.914)	0.826 (0.715–0.936)	0.733 (0.428–1.000)	0.752
ALBI	0.669 (0.529–0.809)	0.762 (0.637–0.887)	0.750 (0.605–0.895)	0.547 (0.287–0.807)	0.659
CPS	0.649 (0.497–0.801)	0.689 (0.550–0.829)	0.684 (0.484–0.782)	0.578 (0.263–0.892)	0.620
MELD	0.602 (0.436–0.767)	0.635 (0.490–0.781)	0.633 (0.484–0.782)	0.429 (0.166–0.693)	0.580
GALAD	0.773 (0.637–0.909)	0.796 (0.678–0.914)	0.810 (0.697–0.923)	0.642 (0.349–0.934)	0.763
ASAP	0.777 (0.642–0.912)	0.812 (0.699–0.924)	0.838 (0.726–0.950)	0.679 (0.315–1.000)	0.774
GAAP	0.785 (0.657–0.913)	0.824 (0.715–0.933)	0.850 (0.740–0.960)	0.689 (0.311–1.000)	0.777
BALAD	0.740 (0.602–0.877)	0.782 (0.663–0.901)	0.788 (0.667–0.909)	0.733 (0.549–0.918)	0.727
BALAD-2	0.796 (0.676–0.916)	0.831 (0.725–0.938)	0.807 (0.686–0.929)	0.637 (0.297–0.976)	0.762
DOYLESTOWN	0.807 (0.687–0.928)	0.814 (0.700–0.928)	0.846 (0.730–0.963)	0.684 (0.301–1.000)	0.770
AMAP	0.554 (0.389–0.719)	0.648 (0.500–0.796)	0.647 (0.500–0.794)	0.538 (0.234–0.841)	0.579
BCLC	0.789 (0.668–0.909)	0.837 (0.726–0.948)	0.774 (0.638–0.910)	0.575 (0.249–0.902)	0.732

AFP: Alpha-fetoprotein; AFP-L3: Lens culnaris agglutinin-reactive alpha-fetoprotein; ALBI: Albumin-bilirubin; BCLC: Barcelona Clinic Liver Cancer; CPS: Child–Pugh Score; DCP: Des-gamma-carboxy prothrombin; MELD: Model-for End Stage Liver Disease.

**Table 5 cancers-17-03457-t005:** Prognostic Performance of Individual Biomarkers and Models by Treatment Category.

**Curative treated (n = 71)**					
	**1st-Year AUROC**	**2nd-Year AUROC**	**3rd-Year AUROC**	**5th-Year AUROC**	**c-İndex**
AFP	0.801 (0.594–1.000)	0.727 (0.566–0.888)	0.613 (0.472–0.754)	0.579 (0.406–0.752)	0.633
AFP-L3	0.792 (0.630–0.953)	0.670 (0.519–0.820)	0.525 (0.383–0.667)	0.533 (0.335–0.732)	0.601
DCP	0.749 (0.574–0.923)	0.786 (0.668–0.905)	0.691 (0.565–0.816)	0.559 (0.379–0.739)	0.654
ALBI	0.676 (0.438–0.914)	0.729 (0.578–0.880)	0.770 (0.656–0.884)	0.569 (0.374–0.765)	0.689
CPS	0.652 (0.401–0.904)	0.661 (0.493–0.829)	0.699 (0.568–0.830)	0.606 (0.425–0.787)	0.616
MELD	0.604 (0.367–0.839)	0.610 (0.460–0.760)	0.636 (0.498–0.775)	0.438 (0.254–0.621)	0.577
GALAD	0.768 (0.547–0.990)	0.769 (0.614–0.925)	0.641 (0.501–0.781)	0.570 (0.386–0.754)	0.652
ASAP	0.755 (0.561–0.949)	0.794 (0.674–0.914)	0.673 (0.542–0.804)	0.568 (0.391–0.746)	0.665
GAAP	0.729 (0.513–0.945)	0.795 (0.664–0.926)	0.691 (0.563–0.820)	0.572 (0.387–0.757)	0.664
BALAD	0.832 (0.691–0.973)	0.737 (0.609–0.864)	0.635 (0.500–0.769)	0.577 (0.379–0.775)	0.652
BALAD-2	0.865 (0.740–0.989)	0.813 (0.709–0.917)	0.709 (0.585–0.833)	0.697 (0.511–0.882)	0.698
DOYLESTOWN	0.729 (0.497–0.962)	0.681 (0.508–0.854)	0.614 (0.474–0.755)	0.602 (0.397–0.8e06)	0.603
AMAP	0.398 (0.136–0.661)	0.652 (0.467–0.838)	0.729 (0.592–0.866)	0.550 (0.368–0.732)	0.636
BCLC	0.784 (0.601–0.967)	0.681 (0.515–0.846)	0.637 (0.504–0.770)	0.414 (0.228–0.600)	0.613
**Non-curative treated ** **(n = 81)**					
	**1st-year AUROC**	**2nd-year AUROC**	**3rd-year AUROC**	**5th-year AUROC**	**c-index**
AFP	0.721 (0.619–0.822)	0.699 (0.602–0.796)	0.653 (0.548–0.757)	0.426 (0.257–0.596)	0.653
AFP-L3	0.695 (0.595–0.796)	0.678 (0.576–0.779)	0.655 (0.545–0.764)	0.643 (0.467–0.818)	0.626
DCP	0.769 (0.676–0.862)	0.744 (0.652–0.836)	0.740 (0.643–0.836)	0.744 (0.387–1.000)	0.687
ALBI	0.699 (0.600–0.798)	0.788 (0.696–0.879)	0.769 (0.664–0.874)	0.516 (0.307–0.725)	0.670
CPS	0.631 (0.523–0.739)	0.693 (0.591–0.795)	0.681 (0.569–0.794)	0.564 (0.151–0.977)	0.618
MELD	0.610 (0.497–0.723)	0.633 (0.527–0.739)	0.582 (0.462–0.701)	0.521 (0.074–0.967)	0.581
GALAD	0.762 (0.667–0.856)	0.758 (0.688–0.848)	0.719 (0.621–0.818)	0.606 (0.238–0.929)	0.687
ASAP	0.799 (0.711–0.887)	0.776 (0.688–0.864)	0.761 (0.666–0.855)	0.612 (0.208–1.000)	0.708
GAAP	0.778 (0.687–0.870)	0.773 (0.685–0.862)	0.751 (0.653–0.848)	0.612 (0.168–1.000)	0.698
BALAD	0.743 (0.648–0.837)	0.779 (0.690–0.868)	0.738 (0.638–0.837)	0.713 (0.540–0.887)	0.679
BALAD-2	0.782 (0.696–0.868)	0.829 (0.749–0.908)	0.784 (0.687–0.880)	0.622 (0.190–1.000)	0.716
DOYLESTOWN	0.757 (0.662–0.868)	0.782 (0.695–0.870)	0.748 (0.648–0.847)	0.506 (0.072–0.941)	0.679
AMAP	0.475 (0.360–0.590)	0.590 (0.482–0.698)	0.523 (0.403–0.642)	0.458 (0.093–0.824)	0.517
BCLC	0.708 (0.610–0.806)	0.679 (0.576–0.782)	0.655 (0.540–0.770)	0.364 (0.000–0.788)	0.626

AFP: Alpha-fetoprotein; AFP-L3: Lens culnaris agglutinin-reactive alpha-fetoprotein; ALBI: Albumin-bilirubin; BCLC: Barcelona Clinic Liver Cancer; CPS: Child–Pugh Score; DCP: Des-gamma-carboxy prothrombin; MELD: Model-for End Stage Liver Disease.

## Data Availability

The data presented in this study are available on request from the corresponding author due to ethical reasons.

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
