# Peer review of "BALAD-2 Emerges as the Most Accurate Prognostic Model in Hepatocellular Carcinoma: Results from a Biobank-Based Cohort Study"

_cancers, 2025, doi:10.3390/cancers17213457_

Round 1
Reviewer 1 Report
Comments and Suggestions for Authors
- What is the definition of HBV cause for HCC? HBsAg(+) or antiHBc(+).
- MASLD may associate with HBV or HCV infection. How can you exclude viral etiology from the number of MASLD in your data?
- Now immunotherapy can be used for first line therapy for advanced HCC. There is no patient using such regimen in your cohort. Would you enroll such patients or discussion in the limitation of the study.
Author Response
We are pleased to resubmit our manuscript (cancers-3919948) entitled ‘’BALAD-2 Emerges as the Most Accurate Prognostic Model in Hepatocellular Carcinoma: Results from a Biobank-Based Cohort Study’’. We found your critique of our initial submission to be very helpful and changed the manuscript in consistent with your suggestions. The modified or added parts are expressed in yellow colour for your recognition. Our point-by-point responses to the critique are outlined below.
Comments and Suggestions for Authors
1. What is the definition of HBV cause for HCC? HBsAg(+) or antiHBc(+).
- Author’s reply: We thank the reviewer for this valuable point. In our study, HBV-related HCC was defined for the presence of hepatitis B surface antigen (HBsAg) positivity. We have clarified this definition in the Materials and Methods section as follow (Page 3, Line 133-135);
- HBV etiology was defined by the presence of hepatitis B surface antigen (HBsAg) positivity, while HCV etiology was defined by the presence of hepatitis C virus antibody (anti-HCV) and/or detectable HCV-RNA.
2. MASLD may associate with HBV or HCV infection. How can you exclude viral etiology from the number of MASLD in your data?
- Author’s reply: We agree with the reviewer’s important observation. To avoid overlap, patients with HbsAg or anti-HCV positivity were classified as viral etiology even if they had metabolic comorbidities and hepatic steatosis on imaging. The MASLD group thus included only those who were negative for both HbsAg and anti-HCV, ensuring exclusion of viral co-infection. We have clarified this approach in the Materials and Methods section as follows (Page 3, Line 135-139);
- Patients with MASLD were identified based on radiologic evidence of hepatic steatosis in the absence of significant alcohol consumption, negativity for both HBsAg and anti-HCV antibodies, and exclusion of other causes of chronic liver disease. In cases where hepatic steatosis co-existed with HBsAg or anti-HCV positivity, the viral etiology was prioritized over MASLD to avoid overlap.
3. Now immunotherapy can be used for first line therapy for advanced HCC. There is no patient using such regimen in your cohort. Would you enroll such patients or discussion in the limitation of the study.
- Author’s reply: We appreciate this timely and relevant comment. Indeed, immune checkpoint inhibitors (ICIs) have become a first-line option for advanced HCC since 2020-2021. However, during the study period (2019-2014), systemic therapy at our center predominantly consisted of sorafenib, as immunotherapy was not yet reimbursed or widely available in Türkiye. We have acknowledged this in the limitations paragraph and noted that future analyses incorporating ICI-treated patients are warranted (Page 13, Line 386-390).
- Another limitation is that none of the patients in this cohort received immune checkpoint inhibitor-based systemic therapy, as such regimens were not yet widely available or reimbursed in Türkiye during the study period (2019-2024). Future studies should include patients treated with modern immunotherapy combinations to further validate biomarker-based prognostic models in this therapeutic context.
Reviewer 2 Report
Comments and Suggestions for Authors
Comments to the Authors
- The manuscript compares prognostic performance across a heterogeneous set of models—some originally validated as prognostic tools (BALAD, BALAD-2) and others developed for early detection or risk stratification (GALAD, ASAP, aMAP, GAAP, Doylestown algorithm). The Introduction should explicitly justify why diagnostic-oriented models are included alongside established prognostic models in a head-to-head performance evaluation. Furthermore, the Discussion notes that “Some studies demonstrated the prognostic utility of GALAD, ASAP, and aMAP models in HCC in comparison to single individual biomarkers,” yet these key references are not cited in the Introduction. I recommend (a) adding a clear rationale for model selection in the Introduction and (b) including the cited prognostic-utility studies of GALAD, ASAP, and aMAP in that same section to align the manuscript’s aims with its supporting evidence.
- Previous studies have suggested that combining BALAD with tumor diameter (S-LAD) improves prognostic accuracy in liver transplant recipients compared to BALAD alone. (World J Gastroenterol 2018 March 28; 24(12): 1321-1331). However, such findings have not been extended to non-transplant populations. The authors should consider evaluating the performance of this combined model in patients who did not undergo liver transplantation.
- The authors state that accurate prediction of survival in patients with HCC is important for guiding treatment choices. However, the study cohort includes patients across various stages and treatment modalities. Given the limited sample size, subgroup analyses by stage or treatment intent may lack sufficient statistical power. The authors should clarify how the findings of this study could be translated into actionable guidance for treatment selection in clinical practice.
- It is unclear whether the cohort includes only patients with newly diagnosed HCC or also those with recurrent disease. This distinction is clinically relevant and should be explicitly stated in the manuscript.
- The authors conclude that “BALAD-2 outperformed other biomarker-based and clinical prognostic models in predicting survival in patients with HCC, particularly in those receiving curative therapy and viral etiologies.” However, the presented data do not sufficiently support a claim of subgroup-specific superiority.
Author Response
We are pleased to resubmit our manuscript (cancers-3919948) entitled ‘’BALAD-2 Emerges as the Most Accurate Prognostic Model in Hepatocellular Carcinoma: Results from a Biobank-Based Cohort Study’’. We found your critique of our initial submission to be very helpful and changed the manuscript in consistent with your suggestions. The modified or added parts are expressed in yellow colour for your recognition. Our point-by-point responses to the critique are outlined below.
Comments and Suggestions for Authors
Comments to the Authors
1. The manuscript compares prognostic performance across a heterogeneous set of models—some originally validated as prognostic tools (BALAD, BALAD-2) and others developed for early detection or risk stratification (GALAD, ASAP, aMAP, GAAP, Doylestown algorithm). The Introduction should explicitly justify why diagnostic-oriented models are included alongside established prognostic models in a head-to-head performance evaluation. Furthermore, the Discussion notes that “Some studies demonstrated the prognostic utility of GALAD, ASAP, and aMAP models in HCC in comparison to single individual biomarkers,” yet these key references are not cited in the Introduction. I recommend (a) adding a clear rationale for model selection in the Introduction and (b) including the cited prognostic-utility studies of GALAD, ASAP, and aMAP in that same section to align the manuscript’s aims with its supporting evidence.
- Author’s reply: We thank the reviewer for this insightful comment. We agree that clarification was needed regarding the rationale for including both diagnostic- and prognostic-oriented models. While GALAD, ASAP, GAAP, aMAP, and the Doylestown algorithm were initially developed for diagnostic or risk stratification purposes, several recent studies have demonstrated their prognostic value in HCC, showing associations with tumor stage, recurrence, and survival outcomes. Our intent was to evaluate whether these widely used biomarker-based scores could also serve prognostic purposes when directly compared to established prognostic models such as BALAD and BALAD-2. We have now added explicit justification for this inclusion in the Introduction part as follows and incorporated the relevant references on their prognostic applications (Page 3, Line 98):
- Although several biomarker-based models such as GALAD, GAAP, ASAP, aMAP and the Doylestown algorithm were originally developed for early detection or HCC risk stratification, emerging evidence indicates that these models also carry prognostic information related to tumor stage, recurrence, and overall survival [19–23]. Given their increasing use in clinical and research settings, a direct head-to-head comparison with established prognostic models such as BALAD and BALAD-2 is warranted to determine their relative predictive accuracy and potential applicability in survival estimation.
- Previous studies have suggested that combining BALAD with tumor diameter (S-LAD) improves prognostic accuracy in liver transplant recipients compared to BALAD alone. (World J Gastroenterol 2018 March 28; 24(12): 1321-1331). However, such findings have not been extended to non-transplant populations. The authors should consider evaluating the performance of this combined model in patients who did not undergo liver transplantation.
- Author’s reply: We appreciate the reviewer’s important suggestion. Unfortunately, our dataset did not include liver transplant recipients, as these patients were excluded from the study per our eligibility criteria. Therefore, we did not directly evaluate the S-LAD model, which integrates tumor diameter as a prognostic modifier to BALAD. Beyond S-LAD, several surgery specific (either transplant or hepatic resection) models have also been proposed integrating tumor diameter to these biomarkers as prognostic tools. However, our study did not focus on therapy specific prognositc models, but included the models that targets all-stage HCCs. Nonetheless, we agree that several tumor-integrated models could have value in the setting of transplant and surgery patients, and we have acknowledged this in the Discussion section as a limitation and added a comment highlighting the potential value of extending such composite approaches (Page 13, Line 390-397)
- While previous studies have proposed combined indices such as LAD, S-LAD, which integrates tumor diameter with the BALAD model to improve post-transplant and post-hepatectomy prognostication, our study did not focus on therapy specific prognostic models, but rather on models applicable across all stages of HCC [33-35]. Future research should evaluate the prognostic performance of BALAD-based models incorporating tumor burden metrics not only in transplant and resection cohort but also in non-curative settings.
3. The authors state that accurate prediction of survival in patients with HCC is important for guiding treatment choices. However, the study cohort includes patients across various stages and treatment modalities. Given the limited sample size, subgroup analyses by stage or treatment intent may lack sufficient statistical power. The authors should clarify how the findings of this study could be translated into actionable guidance for treatment selection in clinical practice.
- Author’s reply: We agree that the prognostic findings should not be overinterpreted as direct treatment-selection tools. The subgroup analyses were exploratory and intended to assess model robustness across treatment categories rather than to derive treatment-specific recommendations. We have clarified this point and rephrased the statement regarding clinical translation to emphasize the supportive (not prescriptive) role of these models in risk stratification and multidisciplinary decision-making in the simple summary (Page 1, Line 25) and discussion (Page 12, Line 374-381) part as follows;
- Accurate prediction of survival in patients with hepatocellular carcinoma (HCC) is important for multidisciplinary decision-making and follow-up.
-
- Although accurate survival prediction is essential for clinical decision-making, our findings are primarily intended to inform risk stratification rather than to serve as stand-alone treatment selection criteria. The subgroup analyses by treatment intent were exploratory and aimed to assess consistency of model performance rather than to provide treatment-specific guidance. In clinical practice, such models can complement established staging systems (e.g., BCLC) by providing individualized survival estimates that aid multidisciplinary discussion and follow-up planning.
4. It is unclear whether the cohort includes only patients with newly diagnosed HCC or also those with recurrent disease. This distinction is clinically relevant and should be explicitly stated in the manuscript.
- Author’s reply: We appreciate this important clarification request. The cohort consisted of exclusively of patients with newly diagnosed, treatment-naïve HCC, confirmed by imaging or histology at initial presentation. Patients with recurrent HCC after prior therapy were excluded. We have added this clarification in the Methods section as follows (Page 3, Line 117-118) ;
- Patients with a newly diagnosed, treatment-naive HCC between January 2019 and January 2024 were included in the analyses.
5. The authors conclude that “BALAD-2 outperformed other biomarker-based and clinical prognostic models in predicting survival in patients with HCC, particularly in those receiving curative therapy and viral etiologies.” However, the presented data do not sufficiently support a claim of subgroup-specific superiority.
- Author’s reply: We appreciate the reviewer’s careful interpretation. We agree that the observed differences across subgroups were modest and exploratory in nature. To avoid overstating the findings, we have revised the conclusion to reflect that BALAD-2 showed consistently higher performance across subgroups rather than definitive superiority. The text in both the Abstract (Page 2, Line 52-54) and Conclusion sections (Page 14, Line 412=420) has been adjusted accordingly as follows;
- BALAD-2 demonstrated consistent and robust prognostic performance compared with other biomarker-based and clinical models across different patient subgroups, particularly among those receiving curative therapy and viral etiologies.
Reviewer 3 Report
Comments and Suggestions for Authors
In the manuscript with the title "BALAD-2 Emerges as the Most Accurate Prognostic Model in Hepatocellular Carcinoma: Results from a Biobank-Based Cohort Study" the authors have analysed several blood biomarkers (AFP, AFP-L3%, DCP, albumin, bilirubin, ALT, ALP), traditional and composite prognostic models/scoring systems (GALAD, BALAD, BALAD-2, GAAP, ASAP, the Doylestown algorithm, aMAP) and their prognostic performance and usefulness in predicting the survival of the hepatocellular carcinoma patients.
The manuscript is interesting. Here are some recommendations for improvement:
- The quality of figures 1 and 2 is currently unsatisfactory. Enlarge the figures and provide a better resolution.
- Are any ROC curves figures available? Such figures would provide more credibility to the written results and tables.
- The last chapter from the Results' section could be placed in the Conclusions' section.
Author Response
We are pleased to resubmit our manuscript (cancers-3919948) entitled ‘’BALAD-2 Emerges as the Most Accurate Prognostic Model in Hepatocellular Carcinoma: Results from a Biobank-Based Cohort Study’’. We found your critique of our initial submission to be very helpful and changed the manuscript in consistent with your suggestions. The modified or added parts are expressed in yellow colour for your recognition. Our point-by-point responses to the critique are outlined below.
Comments and Suggestions for Authors
In the manuscript with the title "BALAD-2 Emerges as the Most Accurate Prognostic Model in Hepatocellular Carcinoma: Results from a Biobank-Based Cohort Study" the authors have analysed several blood biomarkers (AFP, AFP-L3%, DCP, albumin, bilirubin, ALT, ALP), traditional and composite prognostic models/scoring systems (GALAD, BALAD, BALAD-2, GAAP, ASAP, the Doylestown algorithm, aMAP) and their prognostic performance and usefulness in predicting the survival of the hepatocellular carcinoma patients.
The manuscript is interesting. Here are some recommendations for improvement:
1. The quality of figures 1 and 2 is currently unsatisfactory. Enlarge the figures and provide a better resolution.
- Author’s reply: We thank the reviewer for this valuable feedback. We have replaced Figures 1 and 2 with high-resolution, enlarged versions to ensure optimal clarity and readability.
2. Are any ROC curves figures available? Such figures would provide more credibility to the written results and tables.
- Author’s reply: We appreciate the reviewer’s insightful suggestion that visualizing ROC analyses could strengthen the presentation. We generated time-dependent ROC curves for all evaluated models and biomarkers; however, we did not include them in the manuscript because the resulting figures appeared overly complex, with a total of 14 overlapping curves (AFP, AFP-L3, DCP, GALAD, ASAP, GAAP, BALAD, BALAD-2, aMAP, Doylestown, CPS, MELD, ALBI, and BCLC), without providing additional interpretative value beyond the numerical data presented in the tables. Furthermore, time-dependent ROC analyses were performed for each biomarker and model at the 1st, 2nd, 3rd, and 5th years across multiple subgroups (entire cohort, viral etiology, non-viral etiology, curatively treated, and non-curatively treated), resulting in 20 separate ROC analyses and corresponding curves. We are providing an example of the 1-year ROC curve for the entire cohort for illustrative purposes (can be found in the attached word file), but we have intentionally omitted the full set from the revised version to avoid visual overcrowding and potential confusion, as we believe the tabulated data adequately convey the comparative performance of the models.

|
Whole cohort (n=186) |
1st-year AUROC |
|
AFP |
0.766 (0.684-0.848) |
|
AFP-L3 |
0.741 (0.664-0.818) |
|
DCP |
0.794 (0.721-0.866) |
|
ALBI |
0.731 (0.653-0.809) |
|
CPS |
0.654 (0.563-0.744) |
|
MELD |
0.626 (0.529-0.723) |
|
GALAD |
0.804 (0.730-0.878) |
|
ASAP |
0.820 (0.749-0.892) |
|
GAAP |
0.807 (0.735-0.880) |
|
BALAD |
0.801 (0.732-0.871) |
|
BALAD-2 |
0.827 (0.764-0.890) |
|
Doylestown |
0.789 (0.711-0.868) |
|
aMAP |
0.471 (0.374-0.568) |
|
BCLC |
0.788 (0.719-0.857) |
3. The last chapter from the Results' section could be placed in the Conclusions' section.
- Author’s reply: We thank the reviewer for this helpful suggestion. In response, we have moved the summarizing paragraph from the end of the Results section—originally describing the overall prognostic performance of the models—to the Conclusion section. This adjustment improves the logical flow of the manuscript and avoids redundancy. The Results section now concludes with the comparative analyses, while the Conclusion section provides an integrated synthesis of the key findings (Page 14, Line 412-420).
- BALAD-2 demonstrated consistent and robust prognostic performance compared with other biomarker-based and clinical models across different patient subgroups, particularly among those receiving curative therapy and viral etiologies. Among the individual biomarkers, DCP exhibited the highest c-index and AUROCs at all evaluated time points. BALAD-2 provided the strongest overall prognostic discrimination, while the ASAP and GAAP also showed favorable performance. Collectively, these findings support the integration of biomarker based prognostic models —especially BALAD-2— into clinical risk stratification and multidisciplinary decision-making in HCC management.
Round 2
Reviewer 2 Report
Comments and Suggestions for Authors
The authors’ responses to the reviewer’s comments are well-reasoned and appropriate. The manuscript has been sufficiently revised in accordance with the suggestions, and the changes adequately address the concerns raised.